# Specific Methyl-CpG Configurations Define Cell Identity through Gene Expression Regulation

**DOI:** 10.3390/ijms24129951

**Published:** 2023-06-09

**Authors:** Teresa Improda, Valentina Morgera, Maria Vitale, Lorenzo Chiariotti, Fabiana Passaro, Antonia Feola, Antonio Porcellini, Mariella Cuomo, Antonio Pezone

**Affiliations:** 1Dipartimento di Biologia, Complesso Universitario di Monte Sant’Angelo, Università degli Studi di Napoli “Federico II”, 80126 Napoli, Italy; t.improda@gmail.com (T.I.); valemorgera97@gmail.com (V.M.); mariavitale2014@hotmail.com (M.V.); antonia.feola@unina.it (A.F.); antonio.porcellini@unina.it (A.P.); 2Dipartimento di Medicina Molecolare e Biotecnologie Mediche, Università degli Studi di Napoli “Federico II”, 80131 Napoli, Italy; lorenzo.chiariotti@unina.it (L.C.); fabiana.passaro@unina.it (F.P.)

**Keywords:** methyl-CpG, DNA methylation, gene expression, cell identity

## Abstract

Cell identity is determined by the chromatin structure and profiles of gene expression, which are dependent on chromatin accessibility and DNA methylation of the regions critical for gene expression, such as enhancers and promoters. These epigenetic modifications are required for mammalian development and are essential for the establishment and maintenance of the cellular identity. DNA methylation was once thought to be a permanent repressive epigenetic mark, but systematic analyses in various genomic contexts have revealed a more dynamic regulation than previously thought. In fact, both active DNA methylation and demethylation occur during cell fate commitment and terminal differentiation. To link methylation signatures of specific genes to their expression profiles, we determined the methyl-CpG configurations of the promoters of five genes switched on and off during murine postnatal brain differentiation by bisulfite-targeted sequencing. Here, we report the structure of significant, dynamic, and stable methyl-CpG profiles associated with silencing or activation of the expression of genes during neural stem cell and brain postnatal differentiation. Strikingly, these methylation cores mark different mouse brain areas and cell types derived from the same areas during differentiation.

## 1. Introduction

DNA methylation, a covalent alteration of the cytosines in genomic DNA, primarily affects the CpG dinucleotide in vertebrates and is frequently linked to long-term transcriptional repression [1,2]. DNA methyltransferases DNMT1 and DNMT3A/B/L maintain and establish methyl-CpG, respectively [2]. Conversely, methyl-CpG can be eliminated through dilution, passive demethylation by inactivation of DNMTs, or active demethylation by the ten-eleven translocation (TET) and base excision repair (BER) enzymes [3]. However, several recent studies suggest that the roles of methyl-CpG and its oxidized derivatives, including hydroxymethylation (hydroxy-methyl-CpG), are more complex than previously believed [4,5,6,7]. Methyl-CpG can act as a cellular epigenetic trait involved in embryogenesis, cellular differentiation, and reprogramming because of its ability to be passed down through cell divisions. The methyl-CpG profiles can now be examined genome-wide and at a single base resolution thanks to technological advancements. These whole-genome methodologies have been used in studies that have significantly modified our understanding of the genomic distribution and spatio-temporal dynamics of methyl-CpG in several experimental models [8,9]. Our understanding of how methyl-CpG profiles affect genome expression is rapidly developing thanks to the combination of high definition of methyl-CpG profiling and functional studies in various model systems.

In a previous study, we presented a new tool to analyze DNA methylation in complex DNA sequence populations, i.e., MethCoresProfiler. This program identifies and tracks epiallele (alleles different only by methylation) families in complex cell populations such as those derived from the brain or other organs during postnatal differentiation [10]. MethCoresProfiler sorts out populations of DNA sequences derived from specific loci harboring a nucleus or core of CpGs, which are stably methylated in a relevant fraction of sequences in the population. These methylated cores define and mark the epialleles derived from a single ancestor (which generates the epiallele family). With time, CpGs surrounding the CpGs nucleus or core, change the methylation status, because they are not subjected to selection and contribute to the polymorphism and heterogeneity of methylation profiles of all the epialleles present in the population. It is worth noting that in complex mixtures of methylated DNA molecules, the frequency of the individual epialleles is never statistically significant because they are formed by stable (core) and unstable CpGs [10]. In simpler words, the methylation cores define CpGs subjected to selection and characterize a stable core in families of epialleles. The same applies to negatively selected epialleles, which disappear with time in the population of sequences. In both cases, positive and negative selection of epiallele families implies selection on the expression of the gene(s) represented in each family. With the publicly available MethCoresProfiler, we were able to generate high-resolution and dynamic epigenetic maps of specific loci [10], for example, during neurodevelopment. In fact, during the development of the brain, stem-cell fate decisions are tightly correlated with their epigenetic status and expression profiles [11]. To map and identify DNA methylation cores during neurodevelopment, we performed targeting-bisulfite amplicon sequence analysis of five genes involved in murine neural stem-cell differentiation (mESC) to identify and characterize methyl-CpG configurations associated with the specific timing of expression of these genes during mouse brain postnatal differentiation. Our data demonstrate that specific methyl-CpGs cores in the promoters of these genes are closely associated with the induction or activation of transcription during brain differentiation in both the brain areas and isolated cells. We focused our analysis on the promoters, excluding enhancers, because the location of the promoters, not of the enhancers, at the 5′ end of the target gene is invariant.

## 2. Results

### 2.1. Specific Methylated Cores Mark the Promoters of Genes Involved in Murine Postnatal Brain Differentiation

Cellular differentiation is characterized by a reduction in the proliferation potential and expression of markers defining the cell type. The reduction in the developmental potential is driven by epigenetic changes that prevent the risks linked to the expression of non-lineage-related genes in adult cells [12,13,14]. In this context, the specific role of CpG methylation of the regulatory gene regions that control cell differentiation is still debated. Moreover, the methylated segments of differentiation or stemness genes are highly polymorphic and heterogeneous because, when transcribed, they display stochastic hydroxy-methyl-CpGs, which greatly dilute stable methylated and inheritable sequences subjected to selection [10]. Indeed, in several cases where methylation was measured at the single-cell level, a remarkable degree of methylation polymorphism was found in all cell types, with no evidence of clonal and stable epialleles [15,16]. The considerable degree of heterogeneity is also demonstrated by measurements of the entropy index, i.e., the number of individual species within a population of epialleles, according to Shannon [17]. Furthermore, limited sampling and variability in the mean methylation of individual CpGs in the sequence populations introduce additional bias into the analysis. As mentioned above, we solved this problem by extracting common methylation signatures or cores in the population of heterogeneous sequences. In this way, we can identify stable families of epialleles and study their evolution over time, for example, during the differentiation of stem cells [10].

We applied this approach to study the epigenetic evolution of specific genes, induced or repressed, during murine neural stem-cell differentiation (mESC). We analyzed the promoter methylation configuration(s) of gene markers of pluripotency (such as *NANOG*) [18], or of neuronal (*TUBB3*, *DDO*) [19,20] or glial cells (*GFAP*) [21], or of the differentiation or production of brain–derived neurotrophic factor (*BDNF*) [22] at various time points during postnatal mouse brain differentiation. We report here the identification of stable gene-specific methylation signatures that characterize epiallele families of several genes associated with the silencing or induction of the gene expression during mouse brain postnatal differentiation.

#### 2.1.1. *NANOG* Repression Is Associated with De Novo Formation of a Methylated Core in the Promoter

*NANOG* is a key regulator of self-renewal and the maintenance of pluripotency in undifferentiated embryonic stem cells. Each embryonic stem cell (ESC) in a population displays fluctuating Nanog levels that affect cell fate specification. During stem-cell differentiation, *NANOG* is progressively silenced. We examined the methylation status of the *NANOG* promoter (Figure 1A) during ESC differentiation (Figure 1B). We found that all of the cytosines in the region underwent progressive methylation during differentiation with different rates (Figure 1C), generating heterogeneous and polymorphic methylation profiles of the promoter molecules (Figure 1D). Using the tool indicated above (MethCoresProfiler), we identified a stable CpG core tightly associated with progressive *NANOG* silencing in differentiating cells in this heterogenous population of promoter sequences. At T2–4, the *NANOG* promoter shows the first round of methylation at several CpGs, but the stable and statistically relevant methylation core appears and stabilizes at T8–14 (Figure 1E,F). In addition, we note a transient methylation at CpGs 80, which is rapidly substituted by the stable core of CpGs 365–375. With time, the entire promoter region becomes methylated in differentiated cells. We hypothesize that CpGs 80 is OH methylated and with its demethylation at T8, favors the appearance of the core 365–375, which initiates the methylation of the promoter leading to the silencing of the gene (Figure 1E–H).

#### 2.1.2. *TUBB3* Promoter Methylation Does Not Control the Expression during Neuronal Differentiation

*TUBB3* is almost exclusively expressed in neurons and is induced early during the differentiation and reprogramming of neurons [23,24]. Examining the methylation status of the *TUBB3* promoter (Figure 2A), we identified a stable methylated nucleus at CpGs 22–35 (Figure 2E), which was present at the time 0 and did not significantly change over time (Figure 2B,C) during differentiation.

This segment upstream of the gene in mice and humans contains binding sites for several DNA-binding proteins, mainly located on the positive (+) strand [25]. This evidence suggests that *TUBB3* expression is not controlled by DNA methylation of the proximal promoter during neural stem-cell differentiation (Figure 2G), as also confirmed by immunofluorescence (Figure 2H).

#### 2.1.3. BDNF Gene Activation Parallels the Loss of the Methylated Core at the Promoter during Brain Differentiation

Brain-derived neurotrophic factor (*BDNF*) is a member of the neurotrophin growth factor family that stimulates neuron differentiation, maturation, and survival by suppressing apoptosis. *BDNF* also exerts a neuroprotective effect under adverse conditions [26]. We examined the epigenetic configuration of the *BDNF* promoter during differentiation and found a significant reduction in methylation (Figure 3A,B). All CpGs in the promoter were found to be methylated in a small fraction of T0 cells, except CpGs 81–83, which constitute a common methylation core in the *BDNF* promoter present in the majority of undifferentiated cells (T0) (Figure 3). The induction of differentiation was associated with a sharp loss of methylation for all CpGs methylated at a low frequency and of the CpGs 81–83 of the core at T0–T2 (Figure 3C). The loss of methylation reduced the epigenetic heterogeneity and the Shannon entropy during differentiation (Figure 3D). However, our analysis also revealed that the methylated core, CpGs 81–83, present at T0, increased transiently at T2 and decreased at T8–T14 (Figure 3E–G). The 81–83 CpGs were methylated at T0 and T2 in both strands of the *BDNF* promoter, with a slight preference for the (–) strand (Appendix A). The transient methylation rise of 81–83 CpGs in the early phases of differentiation suggests a priming effect of this core on *BDNF* expression induced by differentiation due to OH demethylation.

#### 2.1.4. GFAP Gene Activation Changes the Configuration of the Promoter Methylated Core

Glial fibrillary acidic protein (*GFAP*) is the protein in the astrocytes of the primary intermediate filament (IF). In the human brain, GFAP isoforms display unique expression profiles, which suggest distinct functional roles. One isoform, GFAPδ, is expressed in proliferative radial glia precursors during human brain development. In humans, *GFAP* is a marker of neural stem cells [27]. We examined the methylation profile(s) of the *GFAP* promoter during mouse brain postnatal differentiation. We found that (Figure 4A), during ESC differentiation, all cytosines in the promoter sequence (Figure 4B) at a low frequency undergo methylation (Figure 4C). In this context, we identified a stable methylation core at CpGs 35–84 in undifferentiated cells, which changed its structure at T4 (84–304) and stabilized at T8–T14 (Figure 4E). This nucleus (84–304 CpGs) became dominant in the whole population (Figure 4F) and marked the transcriptional *GFAP* activation (Figure 4G) [28], as also confirmed by immunofluorescence (Figure 4H).

#### 2.1.5. DDO Gene Promoter Methylation Profile and Expression during Postnatal Brain Differentiation

*DDO* (d-aspartate oxidase) levels are induced during brain differentiation and are inhibited in the adult brain, mirroring, reciprocally, d-aspartate levels, which are low in adults [29]. We examined the methylation status of the *DDO* promoter (Figure 5A), which showed a transient increase at T4 and returned to the baseline levels at T8–T14 (Figure 5B). This methylation involved all cytosines in the locus (Figure 5C) and was correlated with a transient reduction in population heterogeneity at T4 (Figure 5D). We identified a methylation core in the *DDO* promoter at position 105–138–150 at T0–T2, which was fully methylated at T4 (105–138–150–226–293–343) and changed configuration at T8–T14 (105–138) (Figure 5E). This nucleus (105–138) significantly increased its frequency in the population up to T4 and then decreased (Figure 5F), marking the activation of the *DDO* transcription (Figure 5G) [30,31].

### 2.2. CpG Methylation Cores Define Cell Identity

Taken together, these data suggest that the methylation cores in the promoter mark the genes that determine the cell identity. To strengthen this hypothesis, we further explored the methylation cores in the region surrounding the promoter of the *DDO* gene in the whole brain during development (accession number PRJEB16320) [30,31]. In the 3Kb region surrounding the *DDO* locus, analyzed by dividing it into seven amplicons, we found changes in methylation and methylated core structures only upstream of the *DDO* promoter (−1000 to +1) (Appendix A) [30]. Moreover, we found the same methylated core (105–138) in region R4, which overlapped with the analyzed region in Figure 5 (Appendix A). To demonstrate that this core was associated with cell identity, we analyzed the methylation status of the *DDO* located in the R4 region in the DNA extracted from brain areas, several cell types isolated from the same brain areas, immortalized A1 neuronal cells, c-myc immortalized neurons at different passages and ESCs upon the induction of neural differentiation [30], and the gut (Figure 6A; see Section 4) [30]. Strikingly, this core and its components mark all brain areas and the cells isolated from the same brain areas [10]. The correlation between the presence of the *DDO* methylation core and the brain areas and/or cells isolated from the same areas is shown in Figure 6B, in which each core component is identified by the CpGs (105–138–150–226–293 relative to the *DDO* transcription start site) and a color code that marks the specific cell type and the mouse brain area(s) analyzed at different times during postnatal differentiation. Specifically, this tree displays several branches that link cells or brain areas to the specific *DDO* core segments, as follows: (1) cells (neural differentiated ES/oligodendrocytes/astrocytes) and cerebellum/prefrontal/striatum area, (red, CpG 105–138), (2) cortex (black, CpG 105–138–150–226–293 and brown, CpG 226–293), (3) cortex/microglia/undifferentiated ES (green, CpG 138–226), and (4) hippocampus (blue, 105–138–150).

In conclusion, the phylogenetic tree in Figure 6 demonstrates that the *DDO* promoter core differentially marks the cells and the specific brain areas during postnatal differentiation [32,33,34,35].

## 3. Discussion

Specific memory mechanisms that regulate gene expression patterns epigenetically have evolved to establish and maintain cellular identity during development. Once determined, these lineage-specific expression profiles must be maintained across cell divisions, defining active or inactive gene expression states [32]. The most noticeable of these marks is the methylation of the carbon-5 of cytosine (5mC), which has traditionally been thought to be incompatible with active transcription when located near or in gene regulatory regions. Furthermore, 5mC can modulate transcription factor binding [33] or induce the binding of specific 5mC-binding proteins, which can lead to the recruitment of co-repressor complexes to methylated target promoters.

The data we have shown demonstrate that the heterogeneity of methylation profiles at the promoter sites of genes expressed or silenced during mouse postnatal brain differentiation is apparent, not substantial, because we identified stretches of 3 or 4 non-contiguous CpGs present in the majority of sequences analyzed. However, considering that in a promoter segment, at low levels, every CpGs is transiently methylated and demethylated during transcription [4,8,10,15,16], every transcribed OH-methyl CpG is detectable as methyl-C when exposed to bisulfite [5]. By extracting significant non–contiguous CpGs methylated from pools of sequences, we identified methylated CpGs that were stable and highly frequent in the population (methylation cores or nuclei). These cores were statistically significant because they were present in molecules non-identical individually, but were derived from the same original precursor (epiallele family) [10]. We identified several epiallele families in the promoters of five genes activated or repressed during postnatal mouse brain differentiation. These families are characterized by cores of 3, 4, and 6 methylated CpGs that are found in various combinations in different cell types and brain areas. Progressive demethylation of these cores in the promoters of several genes involved in mouse brain postnatal differentiation is associated with the activation of the transcription of specific cells and areas of differentiating mouse brain (*BDNF*; Figure 3). The contrary (progressive methylation), as in the *NANOG* promoter, is associated with silencing. However, we also found the presence of a promoter-methylated core (*GAP*) associated with the activation of transcription [21]. The best example of how a methylation core might associate with induction or the timely inhibition of transcription is provided by the *DDO* promoter across 3Kb of a genomic region. The *DDO* promoter displayed a methylation core undergoing transient methylation early during differentiation (T4) followed by demethylation at late stages (T8–T14).

Moreover, we show that differentiated stem cells and neurons expressing *DDO* acquired variants from the same original methylated core (Figure 6A), which appeared as an identity marker of the precursor cells (Figure 6B). It is worth noting that the analysis with the current methods (single methylated CpG, entropy, or average methylation) did not reveal any stable methylation core (Appendix A).

We wish to note that our analysis identified the same neuronal signature in the gut cells (Figure 6). This can be explained by the notion that the enteric nervous system (ENS) is derived from the migratory capacity of the neural crest [34,35].

We are aware that the main limitation of this study is the circumstantial nature of the evidence associated with the cell type identity and methylation cores in neurodevelopment. Although we did not demonstrate that methylation promoter profiles de facto silence or activate transcription, the promoter methylation cores we identified are tightly associated with the dynamic expression profiles of four genes important for developing and differentiating ES and mouse brains.

## 4. Materials and Methods

### 4.1. ESC culture and Differentiation

E14Tg2a (BayGenomics, Berkeley, CA, USA) mouse ESCs were maintained, as described elsewhere [36,37,38]. To induce neural differentiation, ESCs were plated onto gelatin-coated dishes at low density (1 × 10^3^–5 × 10^3^ cells/cm^2^) in the following differentiation medium: knockout DMEM supplemented with 10% KSR, 2 mM glutamine, 100 U/mL penicillin/streptomycin (Invitrogen, Waltham, MA, USA), and 0.1 mM β-mercaptoethanol (Sigma, St Louis, MO, USA).

### 4.2. RT–PCR

Total RNAs were extracted using Trizol (Invitrogen, Waltham, MA, USA). RNA (2 μg/reaction) was reverse-transcribed using M-MuLV reverse transcriptase (Thermo Fisher Scientific, Waltham, MA, USA). The list of primers is reported in Appendix A.

### 4.3. Immunofluorescence

For immunofluorescence analysis, ESCs were fixed, permeabilized, and incubated with primary antibodies and an appropriate secondary antibody, as previously described [39]. The nuclei were counterstained with DAPI (1:5000; Calbiochem, St Louis, MO, USA). The following primary antibodies were used: anti-βIII Tubulin (1:400; Sigma, St Louis, MO, USA), anti-GFAP (1:300; Sigma, St Louis, MO, USA), and anti-Nanog (1:400, Abcam, Cambridge, United Kingdom). Alexa Fluor 594 or 488 secondary antibodies were used (1:400; Thermo Fisher Scientific, Waltham, MA, USA). Cells were visualized using an inverted microscope (Leica Microsystems, Wetzlar, Germany) and the images were captured with a digital camera (DFC365 FX; Leica Microsystems, Wetzlar, Germany) using LAS-AF (Leica Microsystems, Wetzlar, Germany). Confocal images were acquired with a LSM510META microscope (Carl Zeiss GmbH, Oberkochen, Baden-Württemberg, Germany) using LSM510 software version 3.2 (Carl Zeiss GmbH, Oberkochen, Baden-Württokemberg, Germany). After acquisition, the images were color corrected using the brightness, contrast, and color-balance commands applied to every pixel in each image.

### 4.4. DNA Extraction

In accordance with the manufacturer’s recommendations, DNA was prepared using the DNeasy^®^ Blood and Tissue Kit from Qiagen in Hilden, Germany. Thermo Scientific’s NanoDrop 2000 was used to check the DNA quality. A 260/280 absorbance ratio was used to determine its quantity (Invitrogen, Q32850).

### 4.5. Bisulfite Treatment and Amplicon Library Preparation

An EZ DNA Methylation Kit was used to perform the genomic DNA bisulfite treatment (Zymo Research, Irvine, CA, USA) according to the manufacturer’s instructions. An amplicon library was sequenced using an Illumina Miseq Sequencer to determine the DNA methylation levels. The bisulfite conversion rate was estimated to be 98–99%. Appendix A contains a list of primers.

### 4.6. Dataset Description of Figure 6

The data are available on the ENA database (accession number: PRJEB16320) [30].

All animals were derived from the Jackson Laboratory-provided C57BL/6J mice. All animal research was carried out as described in [30]. Whole brains were extracted from mice at various developmental stages, including E15, P0, P7, P14, P21, P30, and P60. From two mice, five brain areas were dissected (prefrontal cortex, cortex, hippocampus, cerebellum, and striatum) (P30).

Cortical neurons were isolated from the brains of C57BL/6J mouse embryos that were 17 days old, as described in [30]. Triturated tissues were plated in a culture medium, and ara-C (10 mM) was added within 48 h of plating to prevent the growth of non–neuronal cells.

Mixed glial cells, purified microglia, oligodendrocytes, and astrocytes were prepared from the primary mouse, as described in [30], obtaining IB4-FITC or OX42-positive cells, NG2-positive cells, and GFAP-positive cells.

Differentiation of embryonic stem cells toward neurons and glia cells was done as previously described [36].

A1 mes c-myc (A1) is a cell line immortalized by infecting a primary mouse mesencephalon-derived cell culture with a c-myc-carrying retroviral vector prepared from 11-day-old embryos (E11), as described in [30].

C57BL/6J mice were also used to obtain gut tissues from three newborn mice (P0 status) and three adult mice (P90 status) [38].

### 4.7. Sequence Handling

Paired-end reads in FASTQ format from the ENA database (accession number: PRJEB16320) and generated in our lab were merged using the PEAR (paired-end read merger) tool, with a minimum overlapping region of 40 nucleotides. Only reads with a mean quality score (Phred) greater than 33 and a length between 400 and 500 nucleotides were kept. The reads were then converted to FASTA format using the PRINSEQ (preprocessing and sequence information) tool. Reads in FASTA format were processed using ampliMethProfiler (available at https://sourceforge.net/projects/amplimethprofiler/, accessed on 5 March 2023) with several quality filters to extract mCpG configurations in single DNA molecules. We only kept reads that were I 50% longer than the reference length, (ii) at least 80% sequence similarity with the primer with the corresponding gene, (iii) at least 98 percent bisulfite efficiency, and (iv) alignment of at least 60% of their bases with the reference sequences. All cytosines in the CpG sequence context had their methylation status coded as methylated (1) or unmethylated (2). Reads with ambiguous CpG dinucleotide calls (including gaps or A or G) were removed. The binary data were analyzed with the MethCoresProfiler [10] in stages.

### 4.8. Statistical Analysis

Average methylation data are expressed as mean ± standard deviation. The unpaired Student t-test was used to make comparisons between two groups. Multiple comparisons were performed using one-way ANOVA and Tukey’s post hoc test. *p*-Values of 10-10 were deemed statistically significant. Pearson’s correlation test was used to assess the relationship between the distribution of epialleles within each stage group. PCA was used to calculate the abundance of each of the 64 epialleles in the analyzed cell population. PC1 explained 31.2% of the observed variance, while PC2 explained 29%. JMP 9 software was used for all statistical analyses (SAS, Cary, NC, USA).

## 5. Conclusions

DNA methylation is an epigenetic modification essential for mammalian development and is crucial for establishing and maintaining cellular identity. Active DNA methylation and demethylation occur during cell fate commitment and terminal differentiation [39,40]. Recent data provide insights into the contribution of DNA methylation to the establishment of epigenetic memory during embryonic development and the modulation of cell-type-specific gene regulatory programs to ensure proper differentiation [41]. Here, we demonstrated that the configurations of methyl-CpGs (methylated nuclei) define cellular identity and correlate with the regulation of the gene expression.

## Figures and Tables

**Figure 1 ijms-24-09951-f001:**
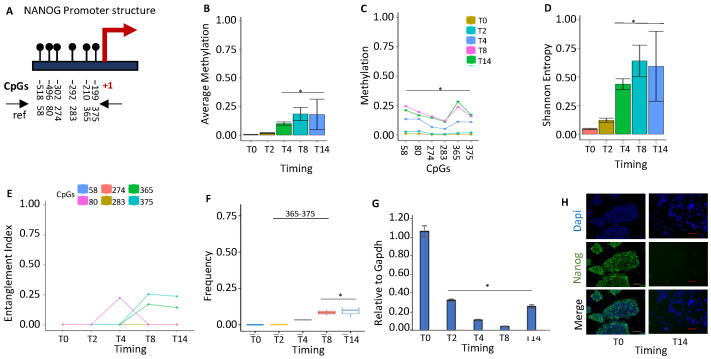
Methylation of *NANOG* promoter during stem-cell differentiation. (**A**) Structure of the *NANOG* promoter. The black lines and circles on the diagram represent each CpG upstream of the transcription start site (TSS). (**B**) Average DNA methylation of the six CpGs shown in (**A**) at various time points (T0, T2, T4, T8, and T14). CpGs are shown as color-coded squares on the right. (**C**) Average CpG methylation is shown in (**A**). (**D**) The methylated molecules’ Shannon entropy of the same samples and time intervals. (**E**) The methylated cores’ composition and structure at various times during differentiation. A color code is used to identify each CpG on the right side of the panel. (**F**) Frequency of the methylated core in the whole population. (**G**) Gene expression analysis. (**H**) Immunofluorescence analysis of Nanog expression in undifferentiated ESCs (T0) and differentiated ESCs (T14). Scale bars: 50 μm. A pairwise comparison was performed with Student’s *t*-test: * *p* < 0.05 versus T0.

**Figure 2 ijms-24-09951-f002:**
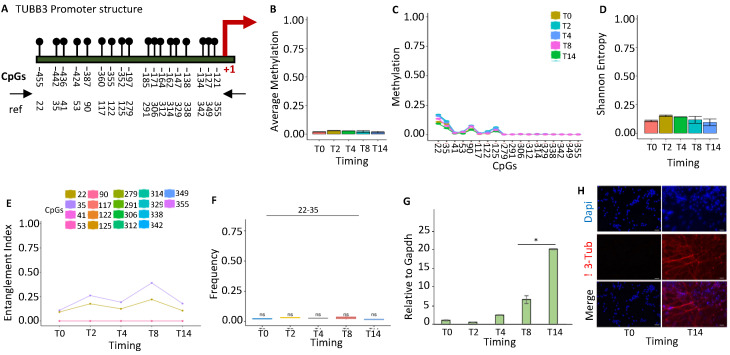
Methylation configuration of the *TUBB3* promoter during stem-cell differentiation. (**A**) Structure of the *TUBB3* promoter. The black lines and circles on the diagram represent each CpG upstream of the transcription start site (TSS). (**B**) Average methylation of the 18 CpGs shown in (**A**) at various stages of stem-cell differentiation (T0, T2, T4, T8, and T14, color-coded squares on the right). Average CpG methylation in the population of molecules, in (**C**). (**D**) Shannon entropy of methylated molecules in the same samples and time intervals. (**E**) The methylated cores’ composition and structure at various times during differentiation. A color code is used to identify each CpG on the right side of the panel. (**F**) Frequency of methylated core in the total population. (**G**) Gene expression analysis. (**H**) Immunofluorescence analysis of TUBB3 expression in undifferentiated (T0) and differentiated ESCs (T14). Scale bars: 50 μm. A pairwise comparison was performed with the student’s *t*-test: * *p* < 0.05 versus T0.

**Figure 3 ijms-24-09951-f003:**
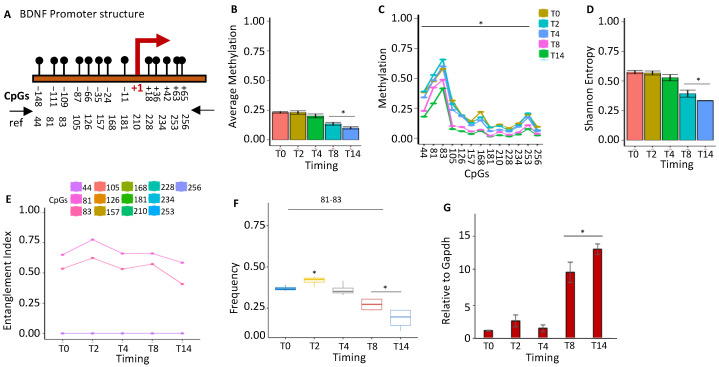
Methylation configuration of the *BDNF* promoter during stem-cell differentiation. (**A**) Structure of the *BDNF* promoter. The CpGs are represented by black circles and lines. (**B**) Average DNA methylation of the 13 CpGs shown in (**A**) at various stages of stem-cell differentiation (T0, T2, T4, T8, and T14, color-coded squares on the right). (**C**) The average level of methylation for each CpG shown in A. (**D**) The methylated molecules’ Shannon entropy for the same samples and time intervals. (**E**) The composition and structure of the methylated cores at various times. A color code is used to identify each CpG on the right side of the panel. (**F**) Frequency of methylated core in the total population. (**G**) Gene expression analysis. A pairwise comparison between each pair was performed with Student’s *t*-test: * *p* < 0.05 versus T0.

**Figure 4 ijms-24-09951-f004:**
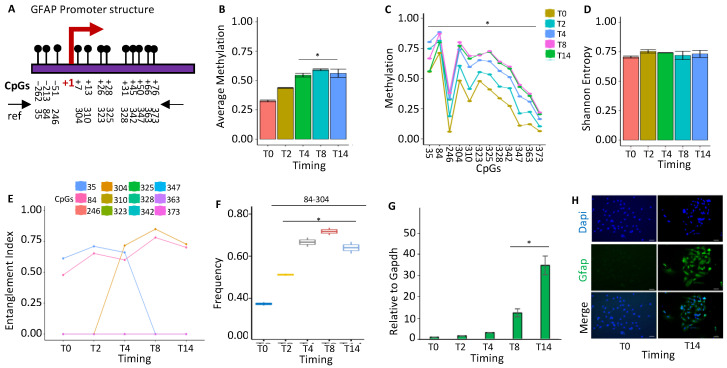
Methylation configuration of *GFAP* promoter during stem-cell differentiation. (**A**) The *GFAP* promoter’s structure. The CpGs are represented by black circles and lines. (**B**) Average methylation of the 12 CpGs depicted in (**A**) in cells at various time points during stem-cell differentiation (T0, T2, T4, T8, and T14, color-coded squares on the right). (**C**) Average CpG methylation as displayed in (**A**). (**D**) The methylated molecules’ Shannon entropy for the same samples and time intervals. (**E**) The composition and structure of the methylated cores at various times. A color code is used to identify each CpG on the right side of the panel. (**F**) Frequency of the methylated core in the total population. (**G**) Gene expression analysis. (**H**) Immunofluorescence analysis of GFAP expression in undifferentiated ESCs (T0) and differentiated ESCs (T14). Scale bars: 50 μm. A pairwise comparison was performed with Student’s *t*-test: * *p* < 0.05 versus T0.

**Figure 5 ijms-24-09951-f005:**
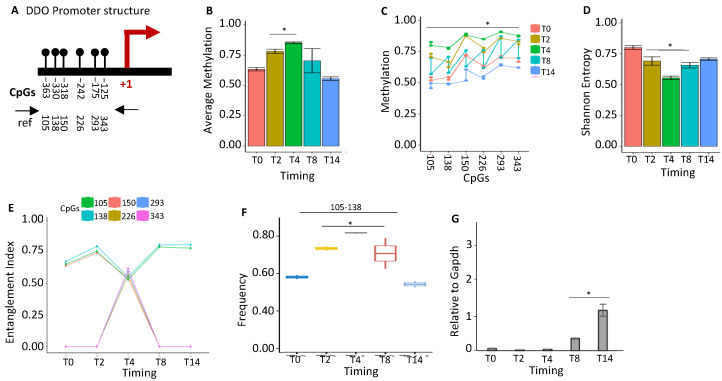
Methylation status of *DDO* promoters during stem-cell differentiation. (**A**) The *DDO* promoter’s structure. CpGs are represented by black circles and lines. (**B**) The average methylation of the six CpGs is shown in (**A**) in the DNA extracted from stem cells at various time points (T0, T2, T4, T8, and T14, color-coded squares on the right). (**C**) The average level of methylation for each CpG found in the populations of molecules, as shown in (**A**). (**D**) The methylated molecules’ Shannon entropy for the same samples and time intervals. (**E**) The composition and structure of the methylated cores at various times. A color code is used to identify each CpG on the right side of the panel. (**F**) Frequency of methylated core in the total population. (**G**) Gene expression analysis. A pairwise comparison was performed with Student’s *t*-test: * *p* < 0.05 versus T0.

**Figure 6 ijms-24-09951-f006:**
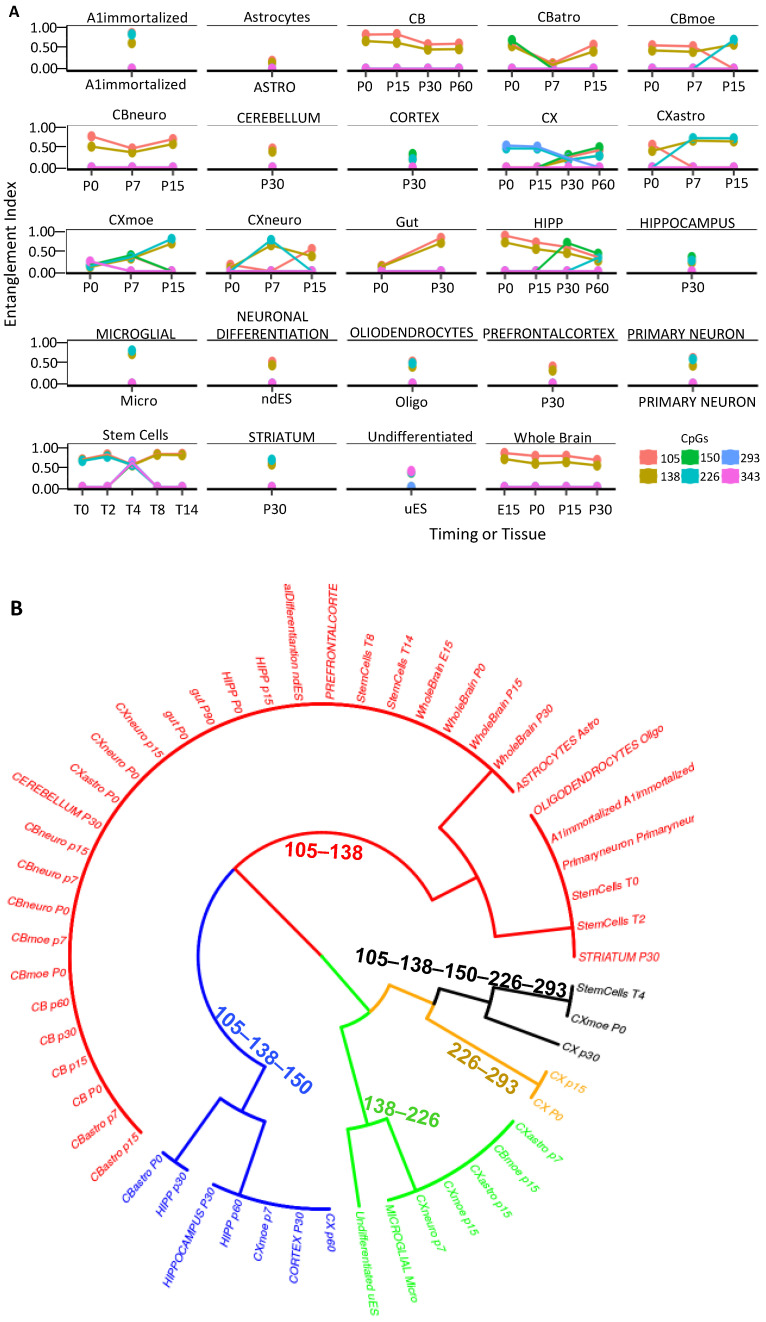
The methyl-CpG configurations of the *DDO* promoter define cell identity and brain areas. (**A**) *DDO* promoter methylated core structure and composition in fractionated cell populations and mouse brain regions at different times (Appendix A). (**B**) A phylogenetic tree describing the relationships between the cell identities and the arrangement of methyl-CpGs. A color code is used to identify each CpG core on the right side of the panel.

## Data Availability

The datasets generated for this study are available from the corresponding author upon request.

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
