# Peer review of "Specific Methyl-CpG Configurations Define Cell Identity through Gene Expression Regulation"

_ijms, 2023, doi:10.3390/ijms24129951_

Round 1

Reviewer 1 Report

Improda and co-authors studied the methylation state of five genes switched on and off during murine postnatal brain differentiation in in vitro system.

Overall, the MS is written in good language and contains the new information in the field.

However, I have some major comments and I could not recommend this manuscript for publication in IJMS in the present form.

Major points:

1.    Why did authors use only promoter regions but not also enhancers? At least in the case of NANOG there is a lot of data about enhancers of this gene.

2.    Authors write about epigenetics hallmarks: Cell identity is determined by chromatin structure and the profiles of gene expression, which are dependent on chromatin accessibility and DNA methylation of regions critical for gene expressions, such as enhancers and promoters.

I completely agree with this. So, authors could include data on the studied genes approximately according to the following plan:

A.   Chromatin structure – DNase or MNase data, or Histone-specific antibodies ChIP, or FAIRE/ATAC assay;

B.   ChIP for most important chromatin modifications: H3K4me1/3, H3K9me, H3K27me3, H3K27ac

C.   X-ChIP with antibodies against most important transcription complexes: PolII, COMPASS, PRC2 (it is well known that PRC2 binds to CpG-islands)

3.    All Figures look like the draft of the figures. Authors should compact the parts and enlarge the font size

4.    Authors used the DDO and GFAP genes as the model for the brain during development (section 2.2). However, there is no RTqPCR for these genes in the current manuscript. In case of DDO authors give a link on the non-published paper: “…marking the activation of the DDO transcription [30,31] (Fabiana Passaro et al, in preparation)”.  However, the RTqPCR data for both genes should be presented in the current manuscript.

Reviewer 2 Report

IJMS-2428173 comments

In this manuscript, the authors collected cells at different time points during embryonic stem cell differentiation. The status of CpG methylation in the promoters of five genes (Nanog, Tubb3, Ddo, Gfap, and Bdnf) was measured by bisulfite DNA Illumina sequencing. The obtained data were evaluated by a MethCoresProfiler program developed in their lab. Finally, they used the same program to analyze the methylation score in the promoter of the DDO gene in the brain during development using the online dataset (accession number PRJEB16320).

The manuscript can be strengthened by addressing the following concerns.

1.        The Method section for the differentiation should be described in more detail. Is it for neuronal differentiation or general differentiation? Is the embryoid body approach applied in the differentiation?

2.        It seems that the differentiation was initiated only by the removal of LIF. If the case, the differentiation may not be completed at the time points as used in the manuscript diagram.

3.        Among the 5 genes selected for the cell identity, four were related to neuronal differentiation. Thus, the authors should use the neuronal differentiation media, instead of just the LIF-deficient media.

4.        Similarly, the expression of neuronal differentiation markers should be measured by quantitative PCR and fluorescent staining. Without the data to show the real cell identity in neuronal differentiation, the “cell identity” by methylation score may not be meaningful at all.

5.        The Method section also lack a brief description for the animal study in Figure 6, even though it was an online dataset. For example, the description for the experimental mice, like strain, age, cell isolation, etc. How were they related to the current study?  

6.        The existing title is too broad. It should be specified to reflect the content as described in the manuscript. At least, it should specify the cell identity at a specific stage or in a specific tissue, e.g embryonic stem cell differentiation or brain development.

     7.   The genes should be presented in italic throughout the manuscript.

Round 2

Reviewer 1 Report

Thanks authors for revised version.

Reviewer 2 Report

I have no more concerns.